# Development and Validation of Confirmatory Foot-and-Mouth Disease Virus Antibody ELISAs to Identify Infected Animals in Vaccinated Populations

**DOI:** 10.3390/v13050914

**Published:** 2021-05-15

**Authors:** Anuj Tewari, Helen Ambrose, Krupali Parekh, Toru Inoue, Javier Guitian, Antonello Di Nardo, David James Paton, Satya Parida

**Affiliations:** 1The Pirbright Institute, Ash Road, Pirbright, Surrey GU24 0NF, UK; anuj474@gmail.com (A.T.); hemarsh@gmail.com (H.A.); krupali.parekh@pirbright.ac.uk (K.P.); antonello.dinardo@pirbright.ac.uk (A.D.N.); david.paton@pirbright.ac.uk (D.J.P.); 2Department of Exotic Disease, National Institute of Animal Health, 6-20-1, Josuihoncho, Kodaira, Tokyo 187-0022, Japan; inouet32@gmail.com; 3Veterinary Epidemiology, Economics and Public Health, The Royal Veterinary College, Hawkshead Lane, North Mymms, Hatfield, Hertfordshire AL9 7TA, UK; j.guitian@rvc.ac.uk

**Keywords:** foot-and-mouth disease, DIVA, NSP ELISA, vaccinate-to-live, sero-surveillance, multiple testing

## Abstract

In foot-and-mouth disease (FMD)-endemic countries, vaccination is commonly used to control the disease, whilst in FMD-free countries, vaccination is considered as an option, in addition to culling the infected and in contact animals. FMD vaccines are mainly comprised of inactivated virions and stimulate protective antibodies to virus structural proteins. In contrast, infection with FMD virus leads to virus replication and additional antibody responses to viral nonstructural proteins (NSP). Therefore, antibodies against NSPs are used to differentiate infection in vaccinated animals (DIVA), in order to estimate the prevalence of infection or its absence. Another advantage of NSP antibody tests is that they detect FMD infection in the field, irrespective of the serotypes of virus in circulation. In cattle, the NSP tests that target the 3ABC polyprotein provides the highest sensitivity, detecting up to 90% of vaccinated animals that become carriers after exposure to infection, with a specificity of around 99%. Due to insufficient diagnostic sensitivity and specificity, detection of a low level of infection is difficult at the population level with a high degree of confidence. The low level of non-specific responses can be overcome by retesting samples scored positive using a second confirmatory test, which should have at least comparable sensitivity to the first test. In this study, six in-house tests were developed incorporating different NSP antigens, and validated using bovine sera from naïve animals, field cases and experimentally vaccinated and/or infected animals. In addition, two (short and long incubation) new commercial NSP tests based on 3ABC competitive blocking ELISAs (ID Screen^®^ FMD NSP Competition, IDvet, France) were validated in this study. The two commercial ELISAs had very similar sensitivities and specificities that were not improved by lengthening the incubation period. Several of the new in-house tests had performance characteristics that were nearly as good as the commercial ELISAs. Finally, the in-house tests were evaluated for use as confirmatory tests following screening with the PrioCHECK^®^ and ID Screen^®^ FMDV NS commercial kits, to assess the diagnostic performance produced by a multiple testing strategy. The in-house tests could be used in series (to confirm) or in parallel (to augment) with the PrioCHECK^®^ and IDvet^®^ FMDV NS commercial kits, in order to improve either the specificity or sensitivity of the overall test system, although this comes at the cost of a reduction in the counterpart (sensitivity/specificity) parameter.

## 1. Introduction

Vaccination is widely used for foot-and-mouth disease (FMD) control in endemic countries. Some countries also use vaccination to maintain freedom, whilst others without FMD virus (FMDV) infection only consider vaccination as an option if the disease is introduced. This form of emergency vaccination may be followed by retention or removal of the vaccinated animals through the adoption of so-called ‘vaccination-to-live’ or ‘vaccination-to-kill’ policies, according to the urgency with which the FMD-free status is to be recovered [1]. FMD vaccination can protect animals against clinical disease and reduce or eliminate virus circulation. However, until virus circulation stops, vaccinated animals can become infected with or without showing clinical disease [2]. An asymptomatic, FMD-persistent infection (the carrier state) can be established in ruminants beyond 28 days post-infection and last up to several years, irrespective of the vaccination status [3], and such animals have been considered a risk for FMD resurgence, even if there is no certainty that this can occur [4]. Furthermore, vaccine effectiveness and vaccine coverage are invariably less than 100%; therefore, in the case of an outbreak controlled by vaccination, sero-surveillance is necessary to substantiate absence of infection and declare freedom from the disease. Vaccines prepared from FMDV antigen that has been purified to remove most viral non-structural proteins (NSP) elicit antibodies mainly against viral structural proteins (SP), whereas infection elicits antibodies against both SP and NSPs. Therefore, NSP tests are used to differentiate infection in vaccinated animals (i.e., DIVA test).

Amongst the FMDV NSPs, 3ABC is considered as the most reliable antigen for DIVA testing [5,6,7,8]. The OIE index method (NCPanaftosa) uses a 3ABC ELISA for initial screening followed by a confirmatory immunoblotting test to detect antibodies against a panel of FMDV NSPs, namely 3A, 3B, 3ABC, 3D, and 2C [1]. However, this testing scheme is laborious and not universally available. Several ELISAs using different NSPs, such as 2B, 2C, 3ABC, 3B, and 3D, have been developed [5,9,10,11,12,13,14,15,16,17,18,19,20]. Among these, one of the most reliable, commercially available tests is the PrioCHECK^®^ FMDV NS (Prionics AG, Switzerland), which is a competitive blocking ELISA [5,8]. Nevertheless, it has been reported that the diagnostic sensitivity of available tests was not sufficient to detect all cases of infection in vaccinated animals [8], whilst imperfect specificity creates difficulties in verifying the free status of herds and flocks. Hence, there remains a need to develop test systems that maximize both sensitivity and specificity—for example, by combining tests and applying simultaneous and/or sequential joint testing schemes [7].

Based on the above assumptions, in this study, six in-house tests were developed incorporating different NSP antigens and these assays were validated using bovine sera from naïve animals, field cases and experimentally vaccinated and/or infected animals. In addition, two (short and long incubation) new commercial NSP tests based on 3ABC competitive blocking ELISAs (ID Screen^®^ FMD NSP Competition, IDvet, France) were validated in this study. Finally, the in-house tests were evaluated for use as confirmatory tests following screening with the PrioCHECK^®^ and ID Screen^®^ FMDV NS commercial kits, to assess the diagnostic performance produced by a multiple testing strategy.

## 2. Materials and Methods

### 2.1. Serum Samples

To determine the diagnostic specificity for each of the new tests, sera collected from 991 naïve Italian cattle were tested. A further collection of 130 sera came from UK cattle involved in trials at the Pirbright Institute but were sampled before they were either vaccinated or experimentally infected. Twenty-one days post-vaccinal sera from these 130 experimental cattle were also used to estimate specificity in a vaccinated population. To determine the diagnostic sensitivity for each of the tests, serum samples that had been derived from several experimental FMD vaccine/challenge studies conducted at the Pirbright Institute over the last 14 years were examined. Experimental bovine sera were derived from four O serotype vaccine-challenge (contact challenge) experiments and five (A, Asia1 and SAT serotypes) European Pharmacopoeia potency tests (needle challenged). The details of these experimental studies have been previously published [2,21,22,23,24]. The experimental sera were classified into four categories according to the vaccination and infection status: unvaccinated–infected–recovered, unvaccinated–infected–carrier, vaccinated–infected–recovered, and vaccinated–infected–carrier (Table 1). The carrier status was determined by whether or not FMDV could be detected by either virus isolation or RT-qPCR from probang samples collected weekly over the period of the experimentation on or after 28 days post-infection. The samples were collected at 35, 56 and 82 days post-challenge (dpc) and included a panel of 36 bovine sera previously established to evaluate NSP tests [23]. A set of 159 field sera from vaccinated and clinically infected Turkish cattle was also tested.

### 2.2. Recombinant Proteins and Peptides Used for Development and Validation of NSP Tests

Four recombinant proteins—3ABC, 3D, 3CD, and 2C—were used to develop and validate the in-house NSP ELISAs. The plasmid constructs pQE-3D and pQE-3CD [25] were kindly provided by Dr. Graham Belsham, Division of Molecular Biology, the Pirbright Institute, and transformed in *E. coli* M15 cells. The previously described plasmid pMF21 expressing FMDV 3ABC [9] was transformed into *E. coli* JM109 cells. Plasmid pMAL-2C was kindly provided by Dr. Jong-Hyeon Park, National Veterinary Research and Quarantine Service, South Korea, and was transformed into *E. coli* JM109 cells. The expression and purification of the protein was performed as per the manufacturer’s instructions. The proteins had different tags allowing purification by affinity chromatography (3ABC: GST tag; 2C: MBP tag; 3D and 3CD: His tags). The 2B peptide used to develop the 2B ELISA test is a peptide of 13 aa in length and has been described by Inoue et al. (2006). The 3B peptide used is a 58 aa full-length peptide. The protein coding sequences of the 3B peptide is as follows: GPYAGPLERQKPLKVRAKLPQQEGPYAGPMERQKPLKVKAKAPVVKEGPYEGPVKKPV.

### 2.3. Optimisation of Antigen and Serum Concentration

All proteins (3ABC, 3D, 3CD, and 2C) were titrated twofold from 8 µg/mL to 0.06 µg/mL in ELISAs. *E. coli* cell lysate was used as a negative control to reduce non-specificity. As for the recombinant proteins, the *E. coli* cell lysate was titrated and optimized for each ELISA. The dilution of serum in each test was also optimized by testing serial dilutions of at least three positive and three negative sera. The *E. coli* expressed total NSPs were quantified using a Bio-Rad Protein Assay kit (Bio-Rad, Hercules, CA, USA) as per the manufacturer’s instruction.

### 2.4. 2C, 3ABC, 3D, and 3CD NS ELISAs

For 3ABC, 3CD and 3D ELISA, the odd-column plates of a 96-well MaxiSorp Nunc-Immuno™ (Sigma, St. Louis, MO, USA) were coated with 4 µg/mL of recombinant antigen and even-column plates were coated with the same amount of *E. coli* antigens as negative controls and incubated at 4 °C overnight. For 2C ELISA, 8 µg/mL of 2C was coated in odd columns and the same amount of *E. coli* antigen was coated in the even columns. For 3ABC and 2C, JM109 cell lysate, and for 3D and 3CD, M15 cell lysate were taken as negative controls as their expression was carried out using these respective *E. coli.* The next day, plates were washed, and sera were added. Known positive- and negative-control sera were included in each test plate. Sera were diluted in blocking buffer (5% marvel, 2% normal rabbit serum and 0.1% tween 20 in PBS) at 1:10 for 3ABC and 2C, 1:16.6 for 3D and 3CD. After serum addition, plates were incubated for 1 h at 37 °C on an orbital shaker, and then washed. Conjugate (anti-bovine IgG) was added at 1:15,000 dilution (optimized by titration) and then incubated for 1 h at 37 °C. Subsequently, plates were washed, and color reaction was developed by adding a chromogen/substrate mixture (50 µL/well) containing 5.05 mM ortho-phenylene-diamine dihydrochloride (Sigma) 30% (*w*/*w*) hydrogen peroxide (Sigma) at 1:2000 dilution. The reaction was stopped after 10 min by the addition of 1 M sulfuric acid and the plates were read on a multi-channel spectrophotometer (Molecular Devices Inc., USA) at 490 nm (A_490_).

### 2.5. 2B/3B Peptide ELISA

The 2B peptide assay had been previously developed at the Pirbright Institute [14]. Although the assay provided a good diagnostic sensitivity, the specificity was not very high. Therefore, the assay was modified by the inclusion of normal horse serum (NHS) in the blocking step (5% marvel, 1% tween 20, 1% NHS in PBS). Briefly, MaxiSorp Nunc-Immuno™ plates were coated with 2B N-cys KLH peptide at 250 ng/mL in carbonate-bicarbonate buffer and kept at 4 °C overnight. Plates were washed three times with PBS and blocked with 200 μL of blocking buffer (as above) for 1 h at 37 °C on an orbital shaker. Subsequently, the plates were washed, and serum samples were added at 1:10 dilution (1% marvel, 1% tween 20, 1% NHS in PBS) in a total volume of 50 μL and then incubated for 1 h at 37 °C on an orbital shaker. Plates were then washed three times and 50 μL of diluted anti-bovine conjugate (1:15,000) was added and incubated for 1 h at 37 °C on an orbital shaker. After the final wash, the colour was developed for spectrophotometer reading as described above. The 3B peptide ELISA was performed as above except that the peptide was coated at a concentration of 200 ng/mL.

### 2.6. Interpretation of Results and Statistical Analysis

For PrioCHECK^®^ (Schlieren-Zuerich, Switzerland) and IDvet^®^ FMDV NS (Rue Louis Pasteur, Grabels, France) tests, the interpretation of their results was performed as per the manufacturer’s protocols. In the PrioCHECK^®^ test, a percentage of inhibition (PI) value of >50% is considered as positive. The test result for each sample for IDvet^®^ was determined as negative-to-positive control ratio (S/N%) = OD sample/OD negative control × 100, thus expressing a percentage of positivity (PP). The diagnostic cut-off for the IDvet^®^ is set at a S/N value of <50%. For the in-house tests, OD values of the positive control (OD_pos_), negative control (OD_neg_) and the test samples (OD_samp_) were corrected by subtracting the OD value of the *E. coli* antigen control (OD_ant_). Finally, the test result (OD_corr_) for each sample was determined as follows:OD(_corr_) = [OD of control or test sample] − [OD of antigen control]

For 2B and 3B peptide ELISAs, the sample OD was taken as the final OD. To determine the performance and diagnostic parameters (specificity and sensitivity) at different cut-off points for each of the in-house NSP ELISAs, non-parametric receiver operating characteristic (ROC) curve analysis was performed in the Stata SE 13 (StataCorp LLC, USA) statistical programme. The reference variable indicating the true state of the observation (infected/non-infected) was extracted from the PCR and/or virus isolation (VI) data on saliva/nasal swabs and oropharyngeal fluid. All FMDV carrier animals were scored as infected. For non-carrier animals, the PCR/VI status from 7dpc to 21dpc was considered. If the non-carrier animal scored positive in PCR/VI between 7dpc and 21dpc, the animal was considered infected; otherwise, it was considered non-infected. For estimating the diagnostic sensitivity and specificity using the field outbreak sera, clinical signs were taken as the reference variable. All animals from field outbreaks that showed clinical signs were considered as infected.

All in-house tests were compared with the PrioCHECK^®^ FMDV NS and IDvet^®^ FMDV NS using both the long and short incubation protocols by comparing their ROC curves, based on the correlated U statistics [26]. The agreement of in-house tests with the PrioCHECK^®^ FMDV NS test and IDvet^®^ long and short incubation FMDV NS tests was expressed in terms of Kappa statistics, and estimating their positive percent agreement (PPA) and negative percent agreement (NPA) [27]. Briefly, Kappa estimates of <0.4, 0.4 to 0.6, 0.6 to 0.8, and >0.8 indicates poor, fair, good, and very good agreement, respectively.

In addition, in order to estimate the diagnostic performance of two correlated tests when used in multiple testing schemes [28], a conditional dependence Bayesian model was developed, as previously described [29]. Two different multiple testing approaches were evaluated: (i) the sampled animals were classified as infected only if both test outcomes were positive (sequential or serial testing, and equivalent to a confirmatory testing)—here, the testing process is aimed at reducing the false positivity rate, thus increasing the specificity; (ii) the animals that test positive in either one or both tests can be classified as infected (simultaneous or parallel testing)—for this case, the goal is to maximize the probability that animals with the disease (true positives) are identified, thus increasing the sensitivity. The diagnostic sensitivity and specificity of multiple testing was evaluated for all groups by combining in-house tests with the PrioCHECK^®^, and IDvet^®^ FMDV NS. Informative priors for both specificity and sensitivity parameters were specified using the non-parametric ROC estimates previously obtained. Calculations were performed in R 3.6.2 [30] using the R2OpenBUGS package to call OpenBUGS 3.2.3 within R [31,32]. For the analysis presented, posterior inferences were based on 50,000 iterations after a burn-in of 5000 iterations were discarded. Convergence was assessed by running three chains from dispersed starting values [33].

## 3. Results

### 3.1. Optimisation of Antigen Concentration

The optimal antigen concentrations for 3ABC, 3D and 3CD assays were found to be 4 µg/mL, and for the 2C test it was 8 µg/mL. The optimal concentration of peptides for the 2B and 3B assays were determined as 250 ng/mL and 200 ng/mL, respectively.

### 3.2. Determination of Cut-Off Values to Estimate Diagnostic Sensitivity and Specificity

Based on the ELISA data from naïve animals, unvaccinated infected and vaccinated and subsequently challenged animals, the cut-off values were determined from non-parametric ROC analyses. The selection of the cut-off value is determined for the purpose for which the assay will be used (determining the prevalence or substantiating the absence of infection) without losing the ability of the test to detect the true status of the test samples. From the ROC analysis, the best trade-off between specificity and sensitivity was estimated at a cut-off point of 0.5 OD, which was thus selected for all in-house ELISAs. The cut-off values for the PrioCHECK^®^ FMDV NS (50% PI) and for the IDvet^®^ FMDV NS (50% S/N%) have already been determined previously by the manufacturers.

### 3.3. Performance of NSP ELISA Tests

The performance of individual in-house tests was estimated from the area under the curve (AUC) and further compared with the PrioCHECK^®^ FMDV NS and two versions (short and long incubation) of the IDvet^®^ FMDV NS (Figure 1). For detecting infection and/or carrier status in unvaccinated infected animals, all the tests, except the 2C, performed very well (AUC range of 0.99) and no significant differences were observed between the individual in-house tests and both the PrioCHECK^®^ FMDV NS and IDvet^®^ FMDV NS tests (*p* > 0.05) (Figure 1, Table 2). Although the 2C test (AUC = 0.85) showed good results for the detection of infection in unvaccinated cattle, the test did not show good performance (AUC = 0.77) in detecting carrier animals (Figure 1, Table 2).

For detecting infection and/or carrier status in vaccinated populations, the performance of all of the in-house tests except 2C were classified as good (AUC range of =0.80 to 0.90) to excellent (AUC range of 0.90 to 0.93) (Figure 1, Table 2). For the vaccinated, needle-challenged animals, the performance of all in-house tests (except 2C) to detect infection and/or carrier was excellent (AUC range of =0.90 to 1.00) and no significant differences were observed between the individual in-house tests and both the PrioCHECK^®^ FMDV NS and IDvet^®^ FMDV NS tests (*p* > 0.05) (data not shown). In contrast, for detecting infection and/or carrier status in vaccinated and contact-challenged animals, the performance of all in-house tests apart from 2C were classified as only fair (AUC range of = 0.70 to 0.80) to good (AUC range of = 0.80 to 0.90) (data not shown).

All in-house tests (except 2C), PrioCHECK^®^ FMDV NS and IDvet^®^ FMDV NS tests were found to have excellent performance when tested with the bovine serum panel for detecting infection (Figure 1, Table 2). For field outbreak sera 2B, 3B and 3ABC, in-house tests gave excellent performance (AUC range of =0.960.99) comparable to that of both the PrioCHECK^®^ FMDV NS and the IDvet^®^ FMDV NS (AUC = 0.99) tests. The 3D and 3CD in-house tests gave good results, returning AUC values of 0.891 and 0.94, respectively (Figure 1, Table 2).

### 3.4. Agreement between NSP ELISA Tests

According to the results provided in Appendix A, all the in-house tests apart from 2C showed a very good agreement with both the PrioCHECK^®^ 3ABC test and IDvet^®^ FMDV NS in testing negative sera for all the categories (i.e., unvaccinated, vaccinated, panel, and field sera), with the percentage correctly classified as negative (NPA) higher than 94%. The percentage of samples correctly classified as positive (PPA) results were found to be different when testing for infection or carrier status: a lower match was obtained when using all the in-house ELISA tests for detecting carrier status in both unvaccinated and vaccinated animals (PPA range of 53.6 to 81.5% and 50.4 to 82.7%, respectively), whilst a higher match was reported when detecting infection in unvaccinated animals (PPA range of 75.5 to 90.4%) (Appendix A). Similarly, low values of PPA were reported for all the in-house tests, apart from 2B and 3B, for detecting infection when testing the bovine serum panel. In samples from known clinically infected cattle from the field, the 2B, 3B and 3ABC in-house tests largely matched the results from both the PrioCHECK^®^ 3ABC test and the IDvet^®^ FMDV NS ELISAs, with PPA values higher than 92.7% (Appendix A). In addition, all the results obtained using the in-house tests were found to a better match to the ones provided by the IDvet^®^ FMDV NS than the PrioCHECK^®^ 3ABC ELISA, with higher PPA values estimated when comparing to the short incubation version (Appendix A).

### 3.5. Detection of Specificity and Sensitivity of NSP Antibody Tests

For all groups of animals, the specificity of all in-house tests, except 2C, ranged from 96.62% to 99.10% (Table 2) at the established cut-off value of 0.5. The specificity of PrioCHECK^®^ FMDV NS and IDvet^®^ FMDV NS tests (long and short incubation) ranged from 99.20% to 99.50% (Table 2). In unvaccinated animals as a whole, the infection and/or carrier status was detected with a sensitivity level of 95.45 to 100% in in-house tests (except 2C), whilst a 100% sensitivity was obtained for both PrioCHECK^®^ FMDV NS and IDvet^®^ FMDV NS tests (long and short incubation) (Table 2). In the vaccinated animals as a whole, the sensitivity of in-house tests (except 2C) to detect infection and/or carrier status was lower (range of 56.49 to 75.97%) in comparison to the commercially available ELISAs, returning values in the range of 70.73 to 90.26% (Table 2). Both unvaccinated and vaccinated animals were further subdivided into contact and needle challenge groups according to their route of infection. For the unvaccinated contact and/or needle challenge groups, all in-house tests performed with a high level of sensitivity (93.75 to 100%) similar to what was estimated for the PrioCHECK^®^ FMDV NS and the IDvet^®^ FMDV NS tests to detect infection and/or carrier status (data not shown). Unlike the detection of infection in unvaccinated contact- and needle-challenged cattle, the sensitivity for the detection of infection and/or carrier status in the vaccinated and contact-challenged animals was lower in the in-house tests than in the PrioCHECK^®^ FMDV NS and the IDvet^®^ FMDV NS tests. However, in the vaccinated and subsequently needle-challenged group, the sensitivity to detect infection using 2B, 3B, 3ABC, and 3D in-house tests was estimated as 82.44%, 90.84%, 80.92%, and 87.02%, respectively, which was closer to the sensitivity of 93.89% and 92.37 % determined for the PrioCHECK^®^ FMDV NS and IDvet^®^ FMDV NS (long) tests, respectively (data not shown). However, sensitivity of detection of infection by 3CD and 2C were only 78.63% and 57.25% for the vaccinated needle-challenged animals. Similarly, the in-house 2B, 3B, 3ABC, and 3D tests performed with a sensitivity of 88.75%, 92.50%, 88.75%, and 85.00%, respectively, to detect vaccinated needle-challenged carriers, which is comparable to the sensitivity (92.50%) reported for both the PrioCHECK^®^ FMDV NS and IDvet^®^ FMDV NS tests (data not shown). In contrast, the sensitivity values obtained for the in-house tests to detect infection and/or carrier status in the vaccinated and contact-challenged group were lower, ranging from 31.08% to 58.11%, whereas the sensitivity for the detection of infection and carrier status using the PrioCHECK^®^ FMDV NS test was estimated as 69.83% and 87.84%, respectively, with the diagnostic sensitivity of the IDvet^®^ FMDV NS estimated as 61.21 to 79.73% (long incubation) and 60.34 to 81.08% (short incubation) (data not shown). Based on the bovine serum panel data, 2B and 3B showed comparable sensitivity (2B = 94.44%; 3B = 88.89%) and specificity (2B = 99.09%; 3B = 98.39%) with the PrioCHECK^®^ FMDV NS (Se = 91.67%; Sp = 99.39%) and IDvet^®^ FMDV NS tests (Se = 91.67%; Sp range of =99.29 to 99.50%) for detecting infection (Table 2). The 3ABC, 3D and 3CD tests reported sensitivity in a range between 72.2% and 75% for detecting infection using the bovine serum panel. Similar diagnostic performances were obtained for testing the field outbreak sera, where three tests (2B, 3B and 3ABC) were found to have comparable sensitivity levels (2B = 96.23%; 3B = 97.48%; 3ABC = 96.23%) to the PrioCHECK^®^ FMDV NS (Se = 96.86%) and IDvet^®^ FMDV NS tests (Se = 97.48% for the short incubation; Se = 100% for the long incubation) (Table 2). The 3D and 3CD in-house tests detected 75.47% and 71.07% of infections, respectively. Although the 2C test detected more than 81% of infected animals, its lower Kappa value (0.37) indicated poor agreement with the commercial ELISA kits and thus was not considered as significant (Appendix A).

### 3.6. Performance of in-House Tests Employed in Multiple Testing Schemes with the PrioCHECK^®^ and IDvet^®^ FMDV NS Tests

Considering all the groups tested, all in-house tests (with the exception of the 2C) showed an improved sensitivity when employed simultaneously (parallel testing) and an improved specificity when applied sequentially (serial testing) with both the PrioCHECK^®^ FMDV NS and IDvet^®^ FMDV NS tests, although at the expense of the other diagnostic parameter (Figure 2, Figure 3 and Figure 4). Similarly, either the sensitivity or the specificity could be increased at the cost of the other parameter compared to use of the PrioCHECK^®^ FMDV NS test or IDvet^®^ FMDV NS tests alone. When using a single commercial ELISA test (either the PrioCHECK^®^ FMDV NS or the IDvet^®^ FMDV NS test) but employed in a serial or parallel diagnostic scheme while changing the cut-off, thus adjusting the sensitivity and specificity of the test accordingly, the obtained diagnostic performance ranges were not as improved as by using different tests based on different NSPs (Appendix A). This was found to be true for all the sample categories.

For all the groups of unvaccinated infected cattle, the sensitivity in parallel testing and the specificity in serial testing reached 100% (Figure 2). For vaccinated infected cattle, the sensitivity by parallel testing and specificity by serial testing were observed in a range of 87 to 93% and 99.9%, respectively, in comparison to 82.49% and 99.39% for the use of the PrioCHECK^®^ FMDV NS test alone, and to 77.33%/74.90% (long/short incubation) and 99.10%/99.30% (long/short) for using the IDvet^®^ FMDV NS test (Figure 3). For the vaccinated contact-challenged group, the sensitivity increased within a range of 73 to 83% when parallel testing was applied, whereas the specificity in serial testing increased to 99.9%, higher than that of the PrioCHECK^®^ FMDV NS (Se = 69.83%, Sp = 99.39%) and the IDvet^®^ FMDV NS (Se = 60.34% for both long and short incubation; Sp = 99.29/99.50% for long/short incubation) used as a single test. Similarly, the diagnostic sensitivity by parallel testing increased up to 94 to 99% and specificity by serial testing increased up to 99.9% in the vaccinated needle-challenged group in comparison to 93.89% and 99.39% for that of the single PrioCHECK^®^ FMDV NS test, and to 92.37%/87.79% (long/short incubation) and 99.39% for that of the IDvet^®^ FMDV NS test (Figure 3).

The sensitivity and specificity for detecting the carrier status in the vaccinated infected carrier group increased within a range of 92 to 97% in parallel testing and 99.9% in serial testing, respectively, in comparison to 90.26% and 99.39% for the PrioCHECK^®^ FMDV NS test, whilst the IDvet^®^ FMDV NS reported sensitivity values of 86.36%/83.77% (long/short incubation) and specificity values of 99.29%/99.50% (long/short incubation) (Figure 3). The sensitivity and specificity for detecting the carrier status in the vaccinated infected contact-challenged group increased to a range of 86 to 94% in parallel testing and 99.9% in serial testing, respectively, in comparison to 87.84% and 99.39% for the PrioCHECK^®^ FMDV NS, and to 60.34% and 99.29%/99.50% (long/short incubation) for the IDvet^®^ FMDV NS test (Figure 3). Similarly, the diagnostic sensitivity and specificity to detect carriers in the vaccinated needle-challenged group increased up to a range of 96 to 99% and 99.9%, respectively, in comparison to 92.5% and 99.39% for the sensitivity and specificity of the PrioCHECK^®^ FMDV NS test, and to 92.37%/87.79% (long/short incubation) and 99.29%/99.50% (long/short incubation) for the sensitivity and specificity estimated by the IDvet^®^ FMDV NS test (Figure 3).

The diagnostic sensitivity and specificity of the PrioCHECK^®^ FMDV NS test (91.67% and 99.39%, respectively) and that of the IDvet^®^ FMDV NS ELISA test (91.67%/88.89% and 99.29%/99.50%, long/short, respectively) could be increased up to 96 to 99% in parallel testing and the specificity up to 99.9% in serial testing with in-house tests when panel sera were used (Figure 4). Similarly, for the field sample analysis, the diagnostic sensitivity and specificity of the PrioCHECK^®^ FMDV NS test (96.86% and 99.39%) and that of the IDvet^®^ FMDV NS test (98.74%/97.48% and 99.29%/99.50%, long/short, respectively) could be increased up to 98 to 99% and 99.9% in parallel and serial testing, respectively, when combined with in-house NSP tests (Figure 4).

## 4. Discussion

Following the economic losses experienced during the UK 2001 FMD outbreak, efforts have been made to facilitate the adoption of a ‘vaccinate-to-live’ control policy, which consists of an emergency ring-vaccination within the surrounding areas of the infected premises followed by sero-surveillance to substantiate the absence of virus carrier animals and of virus circulation and to support a declaration of freedom from FMD infection [1]. Several serological tests can be used to help diagnose FMD and to certify that animal populations or geographical regions are free of FMDV infection. The NSP tests are advantageous as they can detect infection for all FMDV serotypes and can be used to detect infection in vaccinated animals. However, when using these tests to screen for vaccinated animals that have recovered from infection, the accuracy in the detection was not as high as expected [8,34,35].

The study by Brocchi et al. [8] evidenced that tests performed with a diagnostic specificity between 97% and 98% are capable of detecting the status of carriers in vaccinated and subsequently infected cattle with sensitivity values ranging between 68% and 94%. The PrioCHECK^®^ FMDV NS (commercial ELISA) kit was reported to have a sensitivity of 86.4% and performed similarly to the prescribed OIE index method (NCPanaftosa) [8]. It was found that retesting the samples tested as positive by the PrioCHECK^®^ FMDV NS with the same test improved the specificity from 98.1% up to 99.2%, whilst when retesting with the SVANOVIR^TM^ FMDV 3ABC-Ab ELISA, the specificity increased up to 99.98% with a drop in diagnostic sensitivity to 71.2%. Hence, it was suggested to adopt multiple testing strategies to maximize the net diagnostic specificity [7]. It was also recognized that if a more sensitive and specific confirmatory test could be found, then the situation would be further improved. Following this resolution, we have developed, within this study, six new in-house tests and validated them using the well-established PrioCHECK^®^ FMDV NS, along with a new commercial IDvet^®^ FMDV NS test that has similar diagnostic sensitivity and specificity. Furthermore, we have employed, in silico, the six in-house tests and either the PrioCHECK^®^ FMDV NS test or the IDvet^®^ FMDV NS tests within multiple testing strategies, in order to estimate the net diagnostic specificity or sensitivity: these included sequential or serial testing and simultaneous or parallel testing schemes.

In previous studies, the efficacy of the 2B NSP test for detecting infection in sera from cattle under experimental and field conditions has been investigated [14,34,36]. However, it was reported to be associated with some non-specificity, which has now been improved by incorporating normal horse serum in the blocking step. In the present study, four recombinant proteins and two peptides, including the 2B peptide, were evaluated in an indirect-ELISA format and methodically validated using a large collection of negative sera from naïve animals, an extensive and varied number of experimental vaccinated and subsequently infected samples, field outbreak sera from known clinically infected cattle, and a well-established panel of bovine sera [2,21,22,23,24], therefore, capturing the response repertoire of the tests to different system conditions.

In the event of an FMD outbreak, the ability to differentiate infected from vaccinated animals is crucial to regain the FMD-free status [1]. Hence, it is necessary to conduct large-scale post-outbreak serosurveillance to detect NSP antibodies. Particularly in areas characterized by low prevalence, it is essential that the tests should perform with both high sensitivity and specificity, in order to increase the positive predictive value [5]. In FMD-free countries, these NSP assays are used on a herd basis and, therefore, if a single infection is confirmed, then the whole herd is culled, assuming that due to incomplete sensitivity, other unidentified carrier animals could be present. Following the FMD epidemic in the UK in 2001, ~3.5 million sera were tested serologically to demonstrate freedom from infection, and with such a number, even a test with 99% specificity will give rise to ~17,500 false positive reactors. Therefore, a test specificity of >99% is probably required. All in-house NSP assays, except the 2C assay, developed during this study performed with a specificity ranging from 97.2% to 99.2% were classified as having excellent performance characteristics, and also produced comparable results to that of the PrioCHECK^®^ FMDV NS and IDvet^®^ FMDV NS tests for detecting infection and/or carrier status in unvaccinated–infected, vaccinated–needle-challenged animals, field known clinically infected animals, and using the well-characterized bovine serum panel data. As reported in a previous study [8], the detection rate for infected cattle within unvaccinated animals seems to be higher than in vaccinated groups. Amongst the new in-house tests, the 3ABC, 3B and 2B assays produced the highest sensitivity and specificity for the detection of infection and/or carrier status in the unvaccinated–infected and vaccinated–needle-challenged groups. Similar results were obtained for the bovine serum panel, where the 2B and 3B tests (but not the in-house 3ABC test) showed both sensitivity and specificity comparable to the results provided by the PrioCHECK^®^ FMDV NS and the IDvet^®^ FMDV NS tests. In general, all the in-house tests except 2C were classified as performing excellently for detecting infection and/or carrier status in vaccinated–needle-challenged animals. Overall, the 2B, 3B and 3ABC tests showed comparable sensitivity and specificity to the PrioCHECK^®^ and IDvet^®^ FMDV NS tests for detecting infection and/or carrier status in the bovine serum panel, and also in vaccinated–needle-challenged animals. In a similar fashion, the sensitivity and specificity of 2B, 3B and 3ABC was comparable to the PrioCHECK^®^ and IDvet^®^ FMDV NS tests to detect infection in field outbreak sera. However, a low detection rate when testing the vaccinated contact-challenged group was observed for all in-house tests, in contrast with the results obtained for the vaccinated–needle-challenged group, which may be either due to a low replication of the virus in these animals in the presence of antibodies elicited by potent vaccines, or due to a lower level of virus challenge in susceptible contact animals in comparison to animals challenged by the intradermolingual needle route. This was subsequently marked by a low performance to detect infection and/or carrier status for the vaccinated–contact-challenged group in comparison to the unvaccinated–contact-challenged animals, where the infection and/or carrier status were detected with a sensitivity of 100%. Cross-tabulating the dichotomous results extracted from the 3ABC, 2B and 3B tests, a good to very good agreement with the PrioCHECK^®^ FMDV NS and IDvet^®^ FMDV NS was estimated for all groups except for the vaccinated contact-challenged results derived from the 3ABC assay. This would consider all the in-house tests except 2C as producing a good to very good agreement with the PrioCHECK^®^ FMDV NS and IDvet^®^ FMDV NS for detecting infection and/or carrier status in unvaccinated–infected and vaccinated, subsequently needle-challenged groups.

Although all samples for the bovine serum panel were derived from vaccinated and subsequently contact and/or needle-challenged animals, higher sensitivity was observed for the panel in comparison to vaccinated recovered animals, particularly in vaccinated-contact-challenged animals. This could reflect the representative composition of the panel sera, which consists of samples from both vaccinated (nine contact and 15 needle-challenged cattle) and unvaccinated groups (*n* = 12) and hence, a higher detection rate would be expected. The sensitivity and specificity of the 2C assay to detect infection and/or carrier status was low for all groups in both experimental and field conditions. The 2C test, in all cases, was highly non-specific, which could be due to the effect of anti-MBP (maltose binding protein) antibodies present in the bovine serum [37], that could be overcome by the inclusion of a pre-adsorption step, with incubation of sera with MBP protein [16,37,38]. In the 2C ELISA developed earlier [9], 2C was tagged with GST (glutathione S-transferase) and showed a sensitivity and specificity of 94% and 95%, respectively. Similarly, in the 2C protein used in the 2C test developed recently [15] with a His tag protein, a sensitivity of 92.9% and a specificity of 94% was demonstrated. Therefore, the significant use of 2C to detect carriers [39] cannot be disregarded, and our assay needs further improvement either by pre-adsorption of sera with MBP protein or designing a 2C construct of shorter length [13,40,41] to avoid a non-specific response.

The OIE-approved system that is used in South America for serial testing involves a screening by 3ABC ELISA with confirmatory testing against other viral NSPs by western blotting (enzyme linked immuno-electro-transfer blotting or IETB). A test sample is considered positive if antigens 3ABC, 3A, 3B, and 3D (±2C) demonstrate staining densities equal to or higher than that of their appropriate controls. A sample is considered negative if two or more antigens demonstrate densities below their control sera. Test samples not fitting either profile are considered indeterminate [1].

This confirmatory testing undoubtedly improves the specificity of the overall test system, but presumably reduces the sensitivity as well. In addition, the methodology does not lend itself to automation and large-scale use. In this study, we have instead established and evaluated ELISAs for each of the NSPs. This could be a precursor for the development of a multiplex serological assay. Sequentially testing (serial scheme) sera using in-house tests (2B, 3B and 3ABC) along with either the PrioCHECK^®^ FMDV NS or IDvet^®^ FMDV NS increased the specificity, whilst an increase in sensitivity was observed when simultaneously testing sera (parallel scheme). For the parallel testing, a significant increase in sensitivity for detecting infection and/or carrier status was obtained, particularly in the contact-challenged animals. In addition, a clear increase in sensitivity was produced when testing in parallel sera obtained from vaccinated cattle either infected by contact or needle-challenged. The increase in diagnostic performance observed in this study by applying in a serial or parallel scheme two ELISAs based on different recombinant NSPs is higher than potentially using a single PrioCHECK^®^ FMDV NS or IDvet^®^ FMDV NS test and then varying the cut-off according to the requirements. This would suggest that the use of more than one NSP can widen the range of detection of antibodies detected by a serial/parallel system instead of detection of a single NSP using a single test.

It should be considered that the use of multiple testing strategies depends on cost, volume of test, presence and capability of lab infrastructure and, mainly, on the epidemiological system that is investigated. In fact, the gain in diagnostic sensitivity or specificity by means of multiple testing strategies always results in an incidental higher non-specific and non-sensitive response: in simultaneous testing, there is a net gain in sensitivity but a net loss in specificity when compared to either of the tests used individually, increasing the true positive rate at the expense of having more false positives, which is the opposite case for the sequential testing. This needs to be clearly taken into account according to the epidemiological system of investigation. For example, during eradication campaigns or to regain disease-free status, the aim is to substantiate absence of disease and, ideally, the diagnostic test in use should be able to correctly define as negative those animals who are actually free of the disease (i.e., a very low false positive rate with a high negative predictive value). In the face of an outbreak, the priority is to detect and eliminate all infections (and virus circulations) in the system; thus, a test with a very high positive predictive value and a very high sensitivity is required (i.e., reducing the false negative rate). Based on the results presented here, a parallel regime of simultaneously testing can be followed, provided that tests are established with a similarly high level of specificity, whilst a sequential testing (serial regime) would always provide a net diagnostic specificity of 100%.

A PrioCHECK^®^ FMDV NS ELISA and both the long and short incubation IDvet^®^ FMDV NS tests revealed high sensitivity and specificity, and changing the duration of the incubation period had little impact. Using a shorter incubation period has the advantage of a quicker test turnaround, although it may be convenient to have an overnight incubation when testing samples that arrive late in the day. Either of these commercial tests might be used as a screening test and the other one as a confirmatory test to obtain the best sensitivity and specificity. The analysis is ongoing in this regard in our laboratory. Based on the diagnostic parameters estimated here, the 2B, 3ABC and 3B tests can be used as robust and accurate confirmatory DIVA tests along with the PrioCHECK^®^ FMDV NS ELISA or the IDvet^®^ FMDV NS and, therefore, as serosurveillance tools to substantiate the absence of FMD infection. As the screening test is based on the 3ABC protein, a confirmatory test based on a different NSP/peptide (i.e., 2B test) to the screening test would provide further confidence on detection of FMD infection in vaccinated animals (a multiple testing strategy previously applied in the field to confirm recent FMD infections circulating in Cyprus between 2004 and 2005) [36].

## Figures and Tables

**Figure 1 viruses-13-00914-f001:**
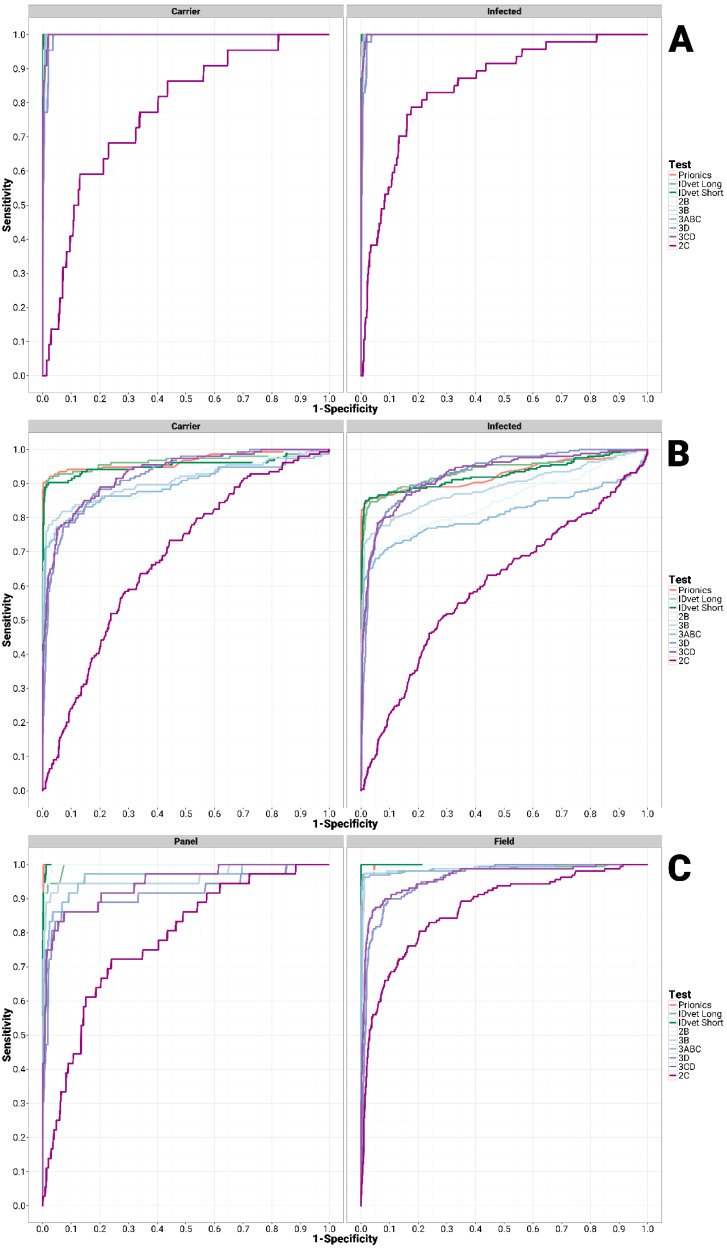
Non-parametric ROC curve comparison of in-house tests with Prionics and IDvet 3ABC tests to detect infection in unvaccinated (**A**), vaccinated (**B**), serum panel (**C**, left) and field sera (**C**, right).

**Figure 2 viruses-13-00914-f002:**
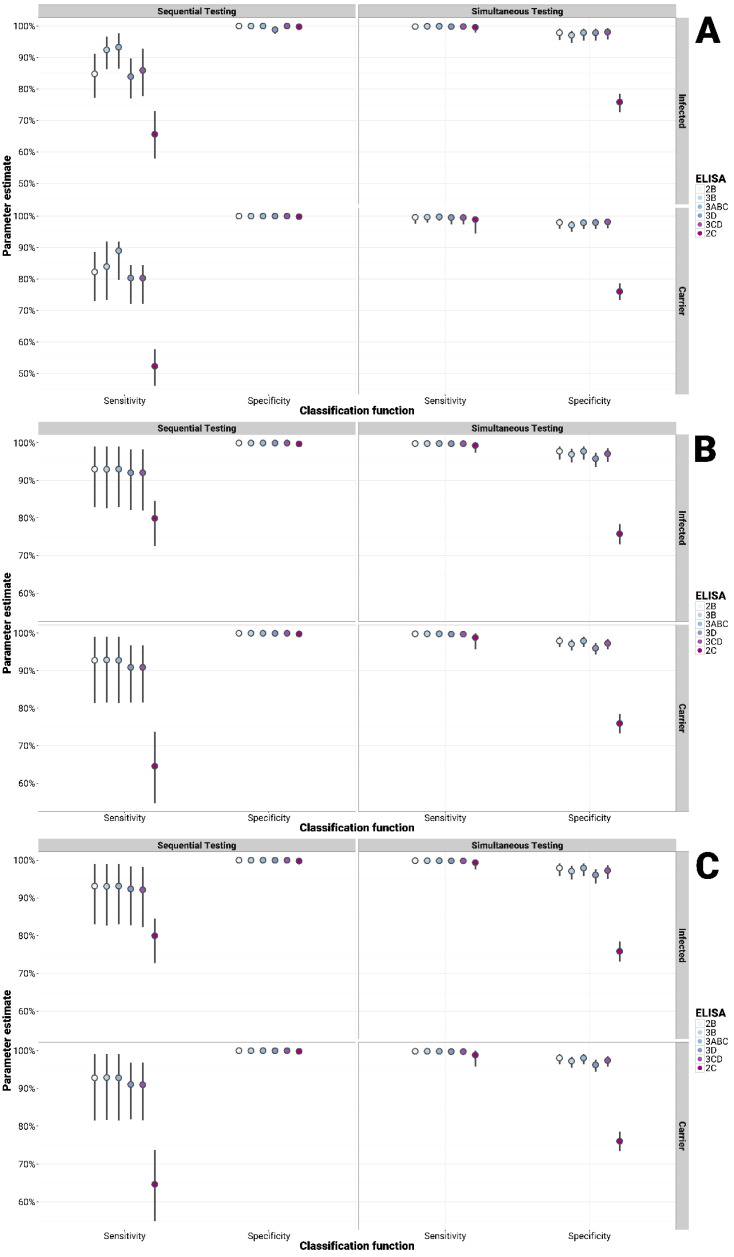
Bayesian analysis for multiple testing of in-house NSP tests with the PrioCHECK^®^ (**A**), IDvet^®^ long (**B**) and short (**C**) incubation FMDV NS tests to detect infection and/or carriers in unvaccinated infected challenged cattle. Serial (sequential testing): samples are classified as either infected and/or carrier only if both test outcomes are positive; parallel (simultaneous testing): samples that test positive in either of the two tests are classified as infected and/or carrier.

**Figure 3 viruses-13-00914-f003:**
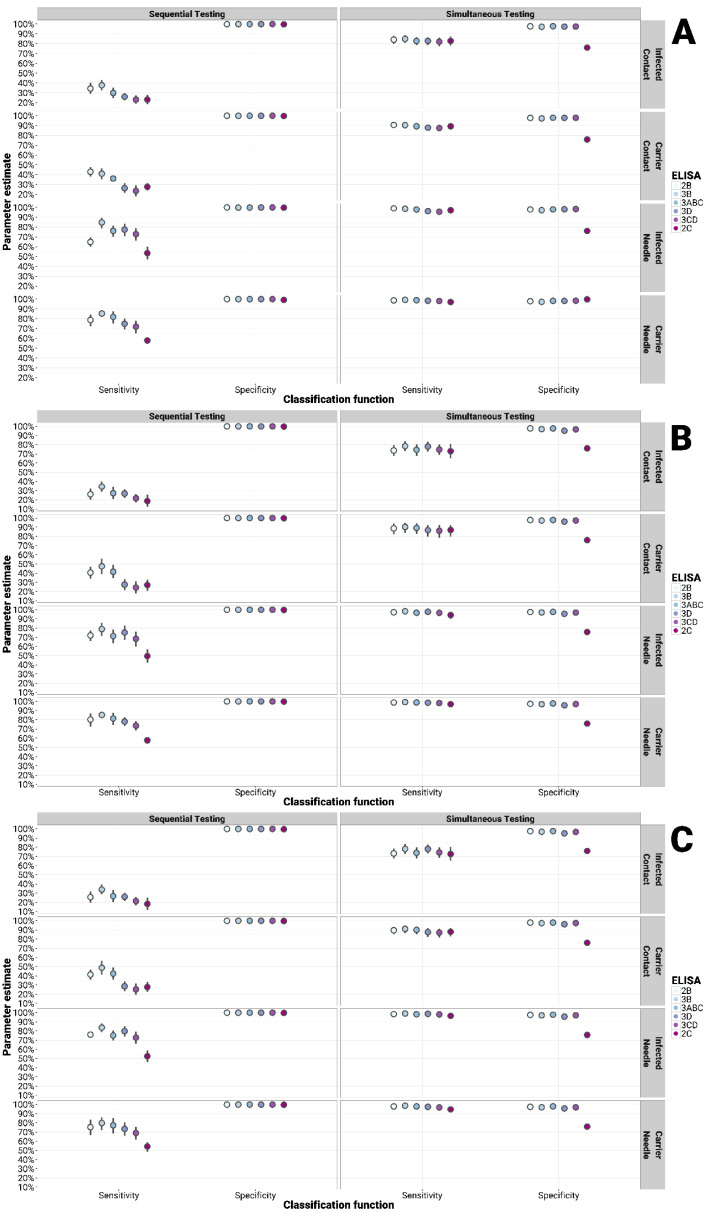
Bayesian analysis for multiple testing of in-house NSP tests with the PrioCHECK^®^ (**A**), IDvet^®^ long (**B**) and short (**C**) incubation FMDV NS tests to detect infection and/or carriers in vaccinated infected challenged cattle. Serial (sequential testing): samples are classified as either infected and/or carrier only if both test outcomes are positive; parallel (simultaneous testing): samples that test positive in either of the two tests are classified as infected and/or carrier.

**Figure 4 viruses-13-00914-f004:**
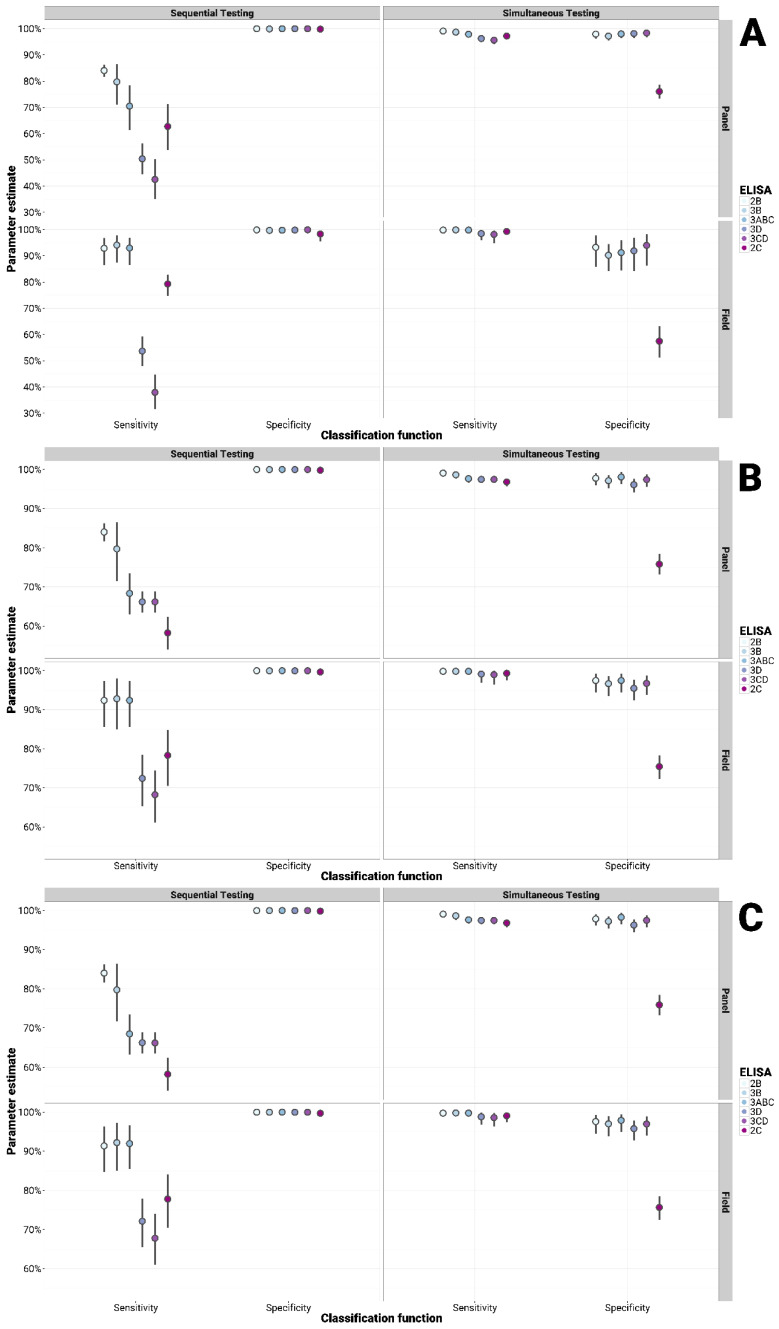
Bayesian analysis for multiple testing of in-house NSP tests with the PrioCHECK^®^ (**A**), IDvet^®^ long (**B**) and short (**C**) incubation FMDV NS test to detect infection in Bovine serum panel and known clinically infected field samples. Serial (sequential testing): samples are classified as either infected and/or carrier only if both test outcomes are positive; parallel (simultaneous testing): samples that test positive in either of the two tests are classified as infected and/or carrier.

**Table 1 viruses-13-00914-t001:** Categorization of the experimental sera used to determine the diagnostic sensitivity for the in-house developed ELISAs, the PrioCHECK^®^ and IDvet^®^ FMDV NS commercial tests.

Group	Infection Route	Number of Animals	Number of Samples
Non-carrier			
	Unvaccinated infected recovered	30	47
	Contact	20	31
	Needle	10	16
	Vaccinated infected recovered	185	261
	Contact	80	130
	Needle	75	131
Carrier			
	Unvaccinated	12	22
	Contact	8	16
	Needle	4	6
	Vaccinated	68	154
	Contact	32	74
	Needle	36	80

**Table 2 viruses-13-00914-t002:** Diagnostic parameters estimated for each of the NSP test for the detection of infection and/or carrier status in unvaccinated, vaccinated, published NSP panel and known clinically infected field sera from cattle. n = number of samples tested (all the categories include the 991 sera from naïve animal of known negative status); AUC = area under the curve; Se = sensitivity (%); Sp = specificity (%).

Test	n	Se	Sp	AUC [95% CI]
Unvaccinated Infected Recovered				
Prionics	1038	100.00	99.39	1.00 [0.99–1.00]
IDvet Long	1038	100.00	99.29	0.99 [0.99–1.00]
IDvet Short	1038	100.00	99.39	0.99 [0.99–1.00]
2B	1038	100.00	99.10	0.99 [0.99–1.00]
3B	1038	100.00	98.39	0.99 [0.99–0.99]
3ABC	1038	100.00	99.09	0.99 [0.99–1.00]
3D	1038	97.87	97.17	0.99 [0.99–0.99]
3CD	1038	97.87	98.59	0.99 [0.99–0.99]
2C	1038	82.98	76.69	0.85 [0.79–0.89]
Unvaccinated Carrier				
Prionics	1013	100.00	99.39	1.00 [0.99–1.00]
IDvet Long	1013	100.00	99.29	0.99 [0.99–1.00]
IDvet Short	1013	100.00	99.39	0.99 [0.99–1.00]
2B	1013	100.00	99.10	0.99 [0.99–1.00]
3B	1013	100.00	98.39	0.99 [0.99–1.00]
3ABC	1013	100.00	99.09	0.99 [0.99–0.99]
3D	1013	95.45	97.17	0.99 [0.98–0.99]
3CD	1013	95.45	98.59	0.99 [0.99–1.00]
2C	1013	68.18	76.69	0.77 [0.68–0.87]
Vaccinated Infected Recovered				
Prionics	1252	82.49	99.39	0.92 [0.90–0.95]
IDvet Long	1252	77.33	99.10	0.93 [0.92–0.95]
IDvet Short	1252	82.49	99.39	0.92 [0.91–0.94]
2B	1252	59.14	99.10	0.83 [0.80–0.87]
3B	1252	72.37	98.39	0.89 [0.86–0.91]
3ABC	1252	59.92	99.09	0.79 [0.75–0.83]
3D	1252	66.15	97.17	0.92 [0.90–0.94]
3CD	1252	57.59	98.59	0.92 [0.90–0.94]
2C	1252	43.19	76.69	0.59 [0.54–0.63]
Vaccinated Carrier				
Prionics	1145	90.26	99.39	0.96 [0.94–0.98]
IDvet Long	1145	86.36	99.29	0.96 [0.95–0.97]
IDvet Short	1145	83.77	99.50	0.95 [0.94–0.96]
2B	1145	70.13	99.10	0.90 [0.87–0.93]
3B	1145	75.97	98.39	0.90 [0.87–0.94]
3ABC	1145	70.78	99.09	0.89 [0.85–0.93]
3D	1145	61.04	97.17	0.92 [0.90–0.94]
3CD	1145	56.49	98.59	0.93 [0.90–0.95]
2C	1145	49.35	76.69	0.68 [0.63–0.72]
Panel				
Prionics	1027	91.67	99.39	0.99 [0.98–1.00]
IDvet Long	1027	91.67	99.29	0.99 [0.99–1.00]
IDvet Short	1027	91.67	99.50	0.99 [0.99–1.00]
2B	1027	91.67	99.10	0.99 [0.99–0.99]
3B	1027	88.89	98.39	0.96 [0.91–1.00]
3ABC	1027	75.00	99.09	0.96 [0.92–1.00]
3D	1027	72.20	97.17	0.91 [0.84–0.98]
3CD	1027	72.20	98.59	0.94 [0.89–0.98]
2C	1027	63.80	76.69	0.78 [0.70–0.85]
Field				
Prionics	159	96.86	99.39	0.99 [0.98–0.00]
IDvet Long	159	100.00	99.29	1.00 [0.99–1.00]
IDvet Short	159	97.48	99.39	0.99 [0.98–1.00]
2B	159	96.23	99.10	0.98 [0.97–1.00]
3B	159	97.48	98.39	0.98 [0.97–0.99]
3ABC	159	96.23	99.09	0.98 [0.97–0.99]
3D	159	75.47	97.17	0.87 [0.83–0.92]
3CD	159	71.07	98.59	0.84 [0.79–0.89]
2C	159	81.76	76.69	0.76 [0.69–0.83]

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
