# Peer review of "Development and Validation of Confirmatory Foot-and-Mouth Disease Virus Antibody ELISAs to Identify Infected Animals in Vaccinated Populations"

_viruses, 2021, doi:10.3390/v13050914_

Round 1
Reviewer 1 Report
1. Over all, this article id too long. Authors are explaining from the basic details. However, it needs to be shorten the length by focusing on the important contents.
2. What is OIE Code in the line 52?
Materials and methods - 2.4 C, 3ABC, 3D, and 3CD NS ELISAs
3. The way of expressing product producers should be unified. Actually, different expressions are used i.e. (Sigma-Aldrich., Ltd., UK), (Sigma), (Molecular Devices Inc., USA), (Schlieren-Zuerich, Switzerland), (Rue Louis Pasteur, GRABELS, FRANCE).
Discussion
Line 415 : Please check the expression 'As per Brocchi et al.(2006)'
Lines 444-445 : please check the expression 'In the event of ~ without vaccinatioin'
Author Response
REVIEWER 1
- Over all, this article id too long. Authors are explaining from the basic details. However, it needs to be shorten the length by focusing on the important contents.
Thank you to the reviewer for this comment and we agree that the manuscript is somewhat lengthy. However, it is very difficult to reduce it further as the manuscript describes the expression of 4 recombinant proteins in different systems and development and validation of 6 new ELISAs including two peptide ELISAs. Further, the results of these 6 ELISAs are compared with two commercial tests (Prionics and IDVet) using various statistical approaches including serial and parallel testing regime. We hope that the reviewer and editor will consider our difficulties for further reducing the length of the manuscript.
- What is OIE Code in the line 52?
This is corrected now by replacing the reference number. It was a reference problem modified as OIE Foot and mouth disease (Infection with foot and mouth disease virus). OIE Terr. Man.2018, 433–464.
3.Materials and methods - 2.4 C, 3ABC, 3D, and 3CD NS ELISAs
2.5 B/3B peptide ELISA
Answer: Corrected to 2.4 2C, 3ABC, 3D and 3CD NS ELISAs
Corrected to 2.5 2B, 3B peptide ELISAs
Track changes made and highlighted in yellow in the manuscript
- The way of expressing product producers should be unified. Actually, different expressions are used i.e. (Sigma-Aldrich., Ltd., UK), (Sigma), (Molecular Devices Inc., USA), (Schlieren-Zuerich, Switzerland), (Rue Louis Pasteur, GRABELS, FRANCE).
Corrected in the manuscript
- Line 415 : Please check the expression 'As per Brocchi et al.(2006)'
We have replaced the entire sentence with: The study of Brocchi et al. [8] evidenced, tests performed with a diagnostic specificity between 97% and 98% are capable to detect thestatus of carriers in vaccinated and subsequently infected cattle with sensitivity values ranging between 68% and 94%.
- Lines 444-445 : please check the expression 'In the event of ~ without vaccinatioin'
We have replaced the sentence with “In the event of an FMD outbreak, the ability to differentiate infected from vaccinated animals is crucial to regain the FMD-free status [1]”
Reviewer 2 Report
Comments
Line 28. ….a second test should have comparable sensitivity, yes, but more important, the two tests should have complimentary sensitivity and specificity. Let us use HIV testing practice in the pre-PCR era as a good example; the ELISA test cutoff could be pushed to maximize the test sensitivity while sacrificing some results being labeled as false positive. This was better than having false negatives. The algorithm would be to confirm all positive results by a more specific test, the Western blot. Please explore the idea that if two tests of complimentary sensitivity and specificity are used, the odds of correctly classifying the correct disease status of the animal/patient is improved.
In general, the paper reads well. There is one major concern in the study design and that is the use of experimental animals to generate data for the analysis of the diagnostic test indices. See the Terrestrial Manual section on this. I think this throws into question the use of the data for the test indice analysis. I think there is no issue of using a kappa analysis to compare the tests, but this issue below needs to be addressed.
According the OIE Terrestrial Manual, Chapter 1.1.6., Stage 2 – Diagnostic Test Performance, pg 81.
2.3 Experimentally infected or vaccinated reference animals
Samples obtained sequentially from experimentally infected or vaccinated animals are useful for determining the kinetics of antibody responses or the presence/absence of antigen or organisms in samples from such animals. However, multiple serially acquired pre- and post-exposure results from individual animals are not acceptable for establishing estimates of DSe and DSp because the statistical requirement of independent observations is violated. Single time-point sampling of individual experimental animals can be acceptable (e.g. one sample randomly chosen from each animal).
Nevertheless it should be noted that for indirect methods of analyte detection, exposure to organisms under experimental conditions, or vaccination, may elicit antibody responses that are not quantitatively and qualitatively typical of natural infection in the target population (Jacobson, 1998). The strain of organism, dose, and route of administration to experimental animals are examples of variables that may introduce error when extrapolating DSe and DSp estimates to the target population. In cases when the near-impossibility of obtaining suitable reference samples from naturally exposed animals necessitates the use of samples from experimental animals for validation studies, the resulting DSe and DSp measures should be considered as less than ideal estimates of the true DSp and DSe.

Author Response
REVIEWER 2
- Line 28. ….a second test should have comparable sensitivity, yes, but more important, the two tests should have complimentary sensitivity and specificity. Let us use HIV testing practice in the pre-PCR era as a good example; the ELISA test cutoff could be pushed to maximize the test sensitivity while sacrificing some results being labeled as false positive. This was better than having false negatives. The algorithm would be to confirm all positive results by a more specific test, the Western blot. Please explore the idea that if two tests of complimentary sensitivity and specificity are used, the odds of correctly classifying the correct disease status of the animal/patient is improved.
The reviewer is right that the confirmatory test is normally applied to increase specificity with its little impact as possible on sensitivity. This is why we have looked at the impact of different test combinations on both sensitivity and specificity (please see Figs 2-4). Furthermore, in the last paragraph of results we have described that “The diagnostic sensitivity and specificity of the PrioCHECK® FMDV NS test (91.67% and 99.39%, respectively) and that of the IDvet® FMDV NS ELISA test (91.67%/88.89% and 99.29%/99.50%, long/short respectively) could be increased up to 96-99% in parallel testing and the specificity up to 99.9% in serial testing with in-house tests when panel sera were used (Fig. 4). Similarly, for the field sample analysis, the diagnostic sensitivity and specificity of PrioCHECK® FMDV NS test (96.86% and 99.39%) and that of the IDvet® FMDV NS test (98.74%/97.48% and 99.29%/99.50%, long/short respectively) could be increased up to 98-99% and 99.9% in parallel and serial testing, respectively, when combined with in-house NSP tests (Figure 4)”.
- In general, the paper reads well. There is one major concern in the study design and that is the use of experimental animals to generate data for the analysis of the diagnostic test indices. See the Terrestrial Manual section on this. I think this throws into question the use of the data for the test indice analysis. I think there is no issue of using a kappa analysis to compare the tests, but this issue below needs to be addressed.
The tests have not only been validated by using experimental and but also field outbreak sera from cattle that were vaccinated earlier and described clearly in the materials and methods. A set of 159 field sera from vaccinated and clinically infected Turkish cattle was tested. Further to determine the diagnostic specificity for each of the new tests, sera were collected from 991 naïve Italian field cattle. Although experimental sera from in-contact and needle challenged animals were used, in most cases, the infection was created by in-contact infected animals to simulate with a field situation. Further the aim of the NSP assay validation is to detect infection in vaccinated animals and also to detect the carrier animals. It is difficult to find out various categories of sera (naïve, vaccinated, vaccinated-infected, unvaccinated-infected, carrier and infected and recovered) in the field and therefore both experimental and field sera have to be used in the study. To meet the 3R situations and maximize the use of animal samples we have used sera before vaccination as naïve sera and then vaccinated sera from the same animal after vaccination, after challenge with live virus as vaccinated-infected sera and then sampled the animals after a long delay to obtain sera from carrier animals. We hope this explains why we have used both experimental and field sera and more than one from some animals as per their status as naïve, vaccinated, vaccinated-infected and carrier (persistently infected).
Round 2
Reviewer 2 Report
I feel the second draft has captured my original concerns and is now suitable for publication.